# Protocolled practice nurse-led care for children with asthma in primary care: protocol for a cluster randomised trial

Sara Bousema,[1] Annemieke J Verwoerd,[1] Lucas M Goossens,[2] Arthur M Bohnen,[1] Patrick J E Bindels,[1] Gijs Elshout[1]

¹General Practice, Erasmus MC, Rotterdam, The Netherlands
²Health Technology, Erasmus School of Health Policy and Management, Rotterdam, The Netherlands

**Correspondence to**
Sara Bousema;
s.bousema@erasmusmc.nl

## ABSTRACT

**Introduction** In children with asthma, daily symptoms and exacerbations have a significant impact on the quality of life of both children and parents. More effective use of asthma medication and, consequently, better asthma control is advocated, since both overtreatment and undertreatment are reported in primary care. Trials in adults suggest that asthma control is better when patients receive a regular medical review. Therefore, protocolled care by the general practitioner may also lead to better asthma control in children. However, such protocolled care by the general practitioner may be time consuming and less feasible. Therefore, this study aims to determine whether protocolled practice nurse-led asthma care for children in primary care provides more effective asthma control than usual care.

**Methods and analysis** The study will be a cluster-randomised open-label trial with an 18-month follow-up. Practice nurses will be the units of randomisation and children with asthma the units of analysis. It is planned to include 180 children aged 6–12 years. Primary outcome will be average asthma control during the 18-month follow-up measured by the Childhood Asthma Control Test (C-ACT). Secondary outcomes include C-ACT scores at t=3, t=6, t=12 and t=18 months; the frequency and severity of exacerbations; cost-effectiveness; quality of life; satisfaction with delivered care; forced expiratory volume in 1 s and forced expiratory flow at 75% and the association of high symptoms scores at baseline and baseline characteristics. Besides, we will conduct identical measurements in a non-randomised sample of children.

**Ethics and dissemination** This will be the first trial to evaluate the effectiveness of protocolled practice nurse-led care for children with asthma in primary care. The results may lead to improvements in asthma care for children and can be directly implemented in revisions of asthma guidelines. The study protocol was approved by the Medical Research Ethics Committee of the Erasmus Medical Centre in Rotterdam.

**Trial registration** NTR6847.

## Strengths and limitations of this study

► This will be the first trial to evaluate the effectiveness of protocolled practice nurse-led care for children with asthma in primary care.
► Asthma in children is a clinical diagnosis for which no 'gold standard' diagnostic criteria are available. 'Physician-diagnosed asthma' will be used as an inclusion criterium in this trial.
► Primary outcome will be asthma control during the 18 month follow-up measured by the Childhood Asthma Control Test. This questionnaire shows good agreement as compared with the Global Initiative for Asthma criteria.

with annual prevalences ranging from 3.0% to 6.5%.[1] A Dutch study on the prevalence of atopic disorders reported that 6.1% of the children aged 0–18 years had asthma.[2] The symptoms include recurrent episodes of wheeze, cough and breathlessness.[3] Asthma in children is associated with significant comorbidity, for example, other airway symptoms and/or (infectious) diseases.[4] Moreover, the symptoms may have a considerable impact on the quality of life of children with asthma and their parents, as well as on healthcare costs. For example, in the Netherlands in 2015, the total costs of airway diseases for children aged ≤15 years were estimated at 312.9 million euros and healthcare costs for primary care for this patient category were 53.6 million euros per year.[5]

### Healthcare system

In the Dutch healthcare system, the general practitioner (GP) has a key role and (almost) everyone is registered with a GP practice. The diagnosis and treatment of asthma, also in children, are part of general practice. In case of diagnostic or treatment problems in children with asthma, referral to secondary care is possible. However, referral to, for example, a paediatrician or paediatric pulmonologist requires prior consent from a GP.

## INTRODUCTION
### Epidemiology and burden of disease
Asthma is the most common chronic disease in children in primary care in the Netherlands

## Treatment

Inhaled corticosteroids (ICS) may be prescribed to reduce the chronic inflammatory condition of the lungs. ICS are the basis of treatment when symptoms are severe enough to justify maintenance treatment, for example, when symptoms occur more than twice a week. Intermittent symptoms of asthma are treated with bronchodilators, starting with a short-acting beta agonist (SABA).[6 7] In some cases of acute severe symptoms of dyspnoea (ie, asthma exacerbation), oral prednisone may be considered.[5]

According to the guideline of the Dutch College of General Practitioners, the goal of asthma treatment is to achieve the best possible asthma control.[6] In case of medication use, the aim is to use the lowest dose/frequency possible to achieve this goal.[6] However, there are strong indications that underdiagnosis and overdiagnosis[8–10] (or too frequent registration of the diagnosis) and undertreatment and overtreatment[11] are present in Dutch general practice.[12] In a large Dutch cohort study, 30% of the children at the age of 8 years with self-reported 'severe current asthma symptoms' were not using ICS (probable undertreatment).[11] Undertreatment can result in unnecessary symptom burden, impaired asthma control and more frequent exacerbations.[13] On the other hand, up to 50% of children with ICS for at least 2 years did not report any wheezing during those 2 years, that is possible overtreatment).[11] Overtreatment may lead to increased healthcare costs and potential iatrogenic effects of the medication.

## Protocolled care and review visits

A recent qualitative study demonstrated that there is a need for an intervention to help parents optimise the management of childhood asthma.[14] Research among children from primary and secondary care diagnosed with moderately severe asthma and treated with ICS showed impaired asthma control at 1-year follow-up and less planned review visits for the children in general practice, as compared with children who were treated in secondary care.[15]

Protocolled care by the GP may lead to better asthma treatment in children. Moreover, it seems to be feasible, since two widely supported evidence-based guidelines are available in the Netherlands for asthma care in children in primary care: (1) the GP guideline for asthma in children developed by the Dutch College of General Practitioners in 2014, which provides guidance for diagnosis and management of (suspected) asthma in children[6] and (2) the care guideline for asthma in children developed by the 'Lung Alliance Netherlands' in 2012.[16] These two guidelines recommend planned reconsultations with structured evaluation of individual care plans and gained goals, allowing to make alterations in the management of the child's asthma more proactively. Unfortunately, in the Netherlands, regular follow-up of children with asthma, as prescribed in these guidelines, is poorly implemented.[12]

International guidelines promote both a policy of optimal symptom control and a focus on prevention of symptom worsening in the subsequent months ('risk factors for poor outcome').[17] With regard to asthma, a systematic review found no significant difference between hospital-based nurse-led care for patients with asthma compared with physician-led care.[18] However, the review included a relatively small number of studies and further research was recommended. Moreover, only two relatively small studies of the (in total) five included studies concerned asthma in children and both of these studies evaluated care by hospital-based specialised asthma nurses; therefore, extrapolation of the results to primary care is insufficiently supported.[19 20]

The results of an Australian cluster randomised controlled trial (RCT) with 195 patients showed that regular spirometry with medical review was associated with improved asthma control in adolescent and adult patients.[21] The authors of a Cochrane review (2003) demonstrated that regular medical review improved healthcare outcomes when combined with education in self-management for adults with asthma.[22] Although regular review visits are needed, this is time consuming and may be less feasible for GPs in primary care.

## Practice nurse-led care

In the Netherlands, the majority of GP practices have practice nurses (otherwise known as general practice-based nurse practitioners or physician assistants; hereafter, referred to as 'practice nurses') whose main task is to perform structured diabetes and cardiovascular risk management and care for patients with chronic obstructive pulmonary disease. This care is supervised by the GPs. In primary care, management of diabetes mellitus can be safely transferred to practice nurses.[23 24] Protocolled care for children with asthma in general practice, supplied by a practice nurse and under supervision of the GP, may give similar (or even better) results in asthma care. However, as shown by a systematic review, there is limited evidence of efficacy for primary care based asthma clinics, and firm conclusions cannot be formed until more good quality trials have been carried out.[25] Furthermore, the (cost-)effectiveness of protocolled asthma management for children in primary care by a trained practice nurse has not yet been evaluated.

## Aim of the study

The aim of this study is to determine the effectiveness of protocolled nurse-led asthma care for children aged 6–12 years in primary care as compared with usual care. The overall aim is to improve asthma care for children in primary care. This will be measured by determining the average asthma control during the 18-month follow-up, as measured by the Childhood Asthma Control Test (C-ACT). Besides improvement for individual children, this may lead to a reduction of the social burden of the disease. Physician-nurse substitution, with a special focus on 'proactive care', may reduce healthcare costs while maintaining quality of care.

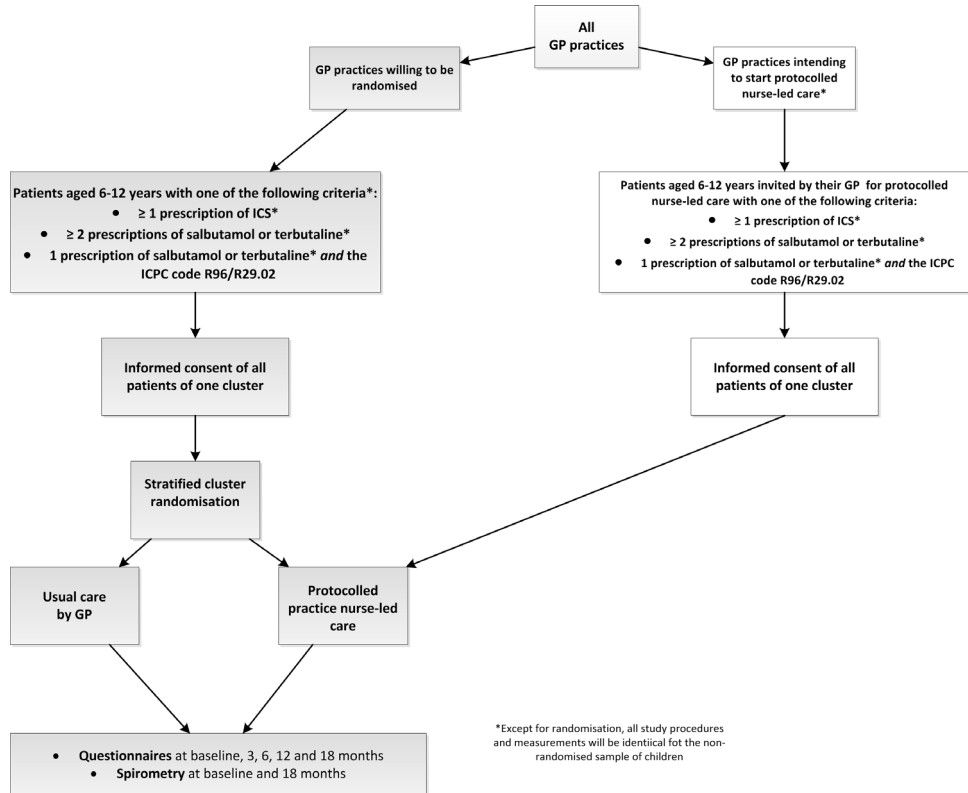

**Figure 1** Flowchart of the study design. GP, general practitioner; ICS, inhaled corticosteroids; ICPC, International Classification of Primary Care.

## METHODS/DESIGN

### Design

The aim is to conduct a cluster RCT with a follow-up of 18 months. Recruited practice nurses (or groups of practice nurses) representing one cluster will be randomised into one of the two arms: (1) protocolled nurse-led care or (2) 'usual care' by the GP. During the study, a number of potential participating GPs were already intending to start protocolled nurse-led care for children with asthma in their practice. Because of this reason, we decided to enable these health centres to participate in the non-randomised part of the trial. This new sample was added after the study started and the change was approved by the ethics committee. All measurements and study procedures (except for the randomisation) will be identical to the randomised group of patients. Figure 1 is a flowchart of the study design.

### Patients

All children aged 6–12 years with asthma registered in the GP practices involved will be invited to participate. Patients are eligible if they fit the following inclusion and exclusion criteria.

### Criteria for inclusion

► Age 6–12 years and with one or more of the following criteria.
► Patients who were prescribed an ICS one or more times in the previous year.

► Patients who were prescribed salbutamol or terbutaline two or more times in the previous year.
► Children with only one prescription of salbutamol or terbutaline in the previous year and a registered International Classification of Primary Care (ICPC) code for asthma (R96) or R29.02 'Prikkelbare Luchtwegen' (bronchial spasm).

### Criteria for exclusion

► Children receiving asthma treatment from secondary care, with secondary care being the main provider of asthma care.
► Children who are not able to perform lung function tests.
► Children with other major chronic diseases which are treated in secondary care; however, children with atopic conditions such as eczema or allergies are not excluded, since these are prevalent comorbidities.
► Children whose parents are unable to understand verbal Dutch instructions or written Dutch questionnaires.

### Primary outcome measure

The primary outcome measure is the average asthma control over 18 months measured by the C-ACT.[26 27]

The C-ACT is a seven-item questionnaire (with a recall window of 4 weeks) that addresses three questions for parents and four questions for children and has been validated in children aged 4–11 years.[27] The C-ACT

questionnaire will be filled out at baseline and at four specified times during the trial. The average C-ACT score will be determined using the measurements on baseline and 3, 6, 12 and 18 months.

## Secondary outcome measures

1. C-ACT outcomes at 3, 6, 12 and 18 months.
2. Frequency and severity of exacerbations. Exacerbations of asthma are episodes characterised by a progressive increase in asthma symptoms, that is, they represent a change from the patient's usual status that is sufficient to require a change in treatment. This will be determined by a questionnaire filled out by the parents.
3. Generic health-related quality of life at 3, 6, 12 and 18 months. This will be measured by the Child Health Utility 9D (CHU9D)[28] and the EuroQol 5D-5L-Y (EQ-5D-Y).[29] Both instruments were specifically designed for use in children.
4. Direct and indirect healthcare costs. The parents will be asked to record healthcare resource use and absence from paid work in diaries. Hospital costs will be assessed by a retrospective chart review at 18 months and by a questionnaire to the parents of the patients.
5. Cost-effectiveness and cost-utility.
6. Disease-specific quality of life measured by the Standardised Paediatric Asthma Quality of Life Questionnaire (PAQLQ(S)) at baseline and at 6 and 18 months.[30]
7. Patient/parent/nurse/GP satisfaction with delivered care. The questionnaire consists of one statement: 'Overall, I am satisfied with current asthma care by my GP practice'. The Likert scale will be as follows: strongly disagree, somewhat disagree, neither agree nor disagree, somewhat agree, or strongly agree.
8. Spirometry: forced expiratory volume in 1 s (FEV1), forced expiratory flow at 75% of vital capacity (FEF 75) and reversibility at baseline and at t=18 months.
9. Medication use in the intervention and control group, measured by a questionnaire.

## Intervention (a protocol for nurse-led care)

For this study, we summarised the guidelines of the Dutch College of General Practitioners and the care guideline for asthma in children, into one concise and easy-to-use protocol for practice nurses. This protocol encompasses an integral assessment of the patient, concerning: patient history, analyses of physical parameters, experienced complaints and restrictions, quality of life, investigation of the possible presence of allergy and determination of the expectations of the patient and their parents with regard to healthcare providers.

In addition, the protocol will contain education about asthma for the child and the parent, guidance on non-pharmacological interventions (eg, lifestyle advice, guidance on physical activity) and guidance on pharmacological interventions (eg, how, which and when different medication can/should be used). All this will lead to an individual care plan for the child. Furthermore, the protocol will contain guidance concerning diagnostic testing, treatment and follow-up. The participating practice nurses will receive up-to-date retraining in protocolled asthma care for children. When new or repeated medication is needed, the GP will approve the prescription.

## Follow-up visits in the intervention group (both randomised and non-randomised)

At baseline and at each control visit, asthma control will be determined by the practice nurse using a four-item questionnaire based on the Global Initiative for Asthma (GINA) guidelines.[17]

The content and frequency of follow-up visits will depend on the asthma control and medication use.

## Follow-up schedule for well-controlled patients in the intervention group

For well-controlled patients (based on GINA criteria) with asthma diagnosed in the previous year, control visits will be planned according to the medication used. For 'as needed' use of SABA, visits will be once in the first 3–6 months. When a patient is well controlled and with asthma diagnosis of >1 year, the control visit will be scheduled annually. When ICS dosage reduction in well-controlled patients is attempted (after 1 year of well-controlled asthma), control visits will be scheduled once every 3 months; at the lowest effective dose once every 3–6 months, and after 1 year, annually.

## Follow-up schedule for partly controlled/uncontrolled patients in the intervention group

Patients with partly or poorly controlled asthma (based on GINA criteria) who do not have optimal medication treatment, will receive additional medication according to the Dutch guideline (ie, add an ICS, or increase the dosage) and will be scheduled for control visits every 2–4 weeks until the asthma is well controlled or optimal medication treatment is reached. In the starting phase of ICS, control visits will be scheduled every 2–4 weeks. When the ICS is continued, review visits take place once in every 3 months in the first year. Children that remain uncontrolled or partly controlled with optimal medication treatment will be referred to secondary care. The referred children will still receive follow-up in the present study; the parents will be requested to continue filling out the electronic questionnaires and spirometry at t=18 months.

Figure 2 is a flowchart of the therapy and follow-up schedule by the practice nurse based on the guidelines of both the Dutch College of General Practitioners[6] and the care guideline for asthma in children.[16]

## Follow-up schedule for the control group

The content and frequency of the control visits for the control group ('usual care') will be determined according to the discretion of the GP.

## Setting

The study will take place in academic and non-academic GP practices in the south-west region of the Netherlands.

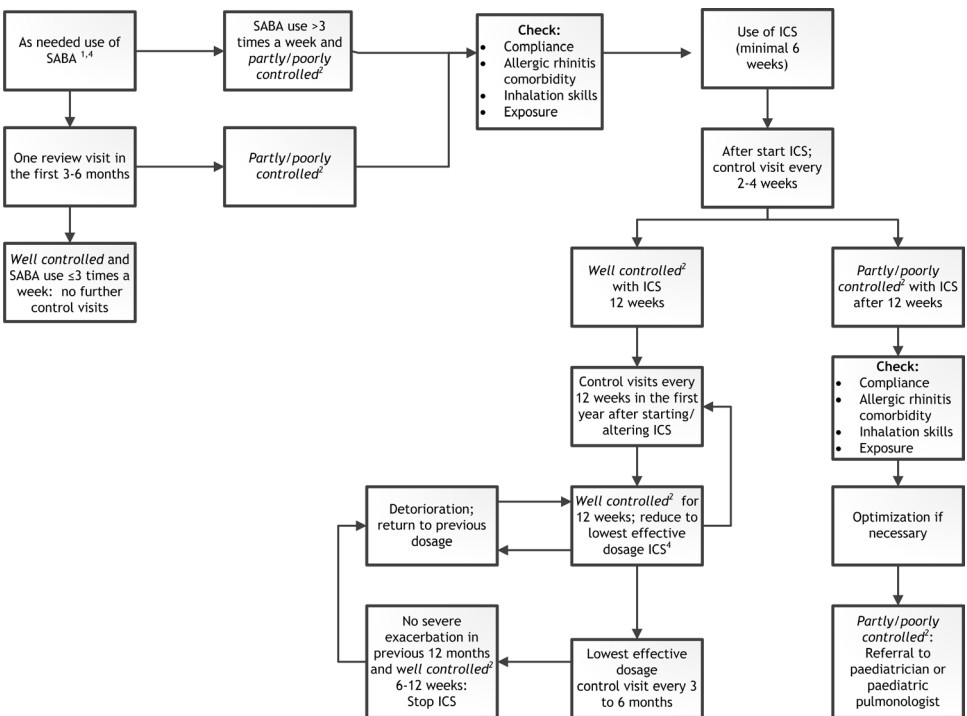

**Figure 2** Flowchart of the therapy and follow-up schedule by the practice nurse. ICS, inhaled corticosteroids; SABA, short-acting beta agonist.

The academic network of GPs 'PrimEUR' in Rotterdam and the south-west of the Netherlands consists of 13 GP health centres with a total of 150 000 registered patients, including ±9000 children aged 6–12 years. This academic network aims to excel in training physicians to be GPs, to supervise medical students during their internships, to initiate healthcare innovation and to participate in research in primary care. Moreover, at least 14 (smaller) GP practices will be recruited to participate in the study. The number of participating GP practices can be enlarged by inviting more GPs to participate.

### Randomisation

The participating health centres (ie, one or more collaborating GP practices) will be the units of randomisation. They will be randomly assigned to either the intervention or control group ('usual care') with a 1:1 allocation as per a computer-generated stratified randomisation schedule. When a practice nurse works for several GP practices, all these practices will be part of the same cluster and randomised as one group. When a health centre consists of several GP practices with their own practice nurse, this health centre will also be considered as one cluster. When two practice nurses are working in one GP practice (not separated), this GP practice or health centre will be randomised as one cluster. When patients are eligible for inclusion in the trial and provide informed consent, a researcher will conduct the randomisation using an online randomisation programme and the patients will be informed in which treatment arm their GP practice is randomised. The cluster randomisation is conducted within 8 weeks after inclusion of the first patient of that GP

practice into the present study. All subsequent patients of the same cluster will receive the same intervention.

Stratification will be based on the size of the GP practice and the number of years of experience of the GP. For example, GP practices in a single health centre (the same postal code) will be randomised in a stratified randomisation of four groups and, therefore, evenly distributed to the intervention and the control arm to minimise differences in baseline characteristics of the patients. A block randomisation with varying blocks (ie, 2–6 blocks) will be used to equally balance both groups. The sequence is kept secret from all researchers involved in the study.

### Sample size

To our knowledge, no previous study has determined the minimally important difference (MID) for the C-ACT for children with asthma in primary care. Therefore, we decided to extrapolate the C-ACT score of a previous study (that included 166 children aged 6–11 years with asthma from secondary care)[26] to our sample size calculation.[23]

Considering an MID of 2, a mean C-ACT score of 19.8 and with a SD of 4.1,[26] a sample size of 126 children will be needed to achieve a power of 80% with an alpha of 5%. Using an intracluster correlation of 0.04[31] and inflating the sample size with 15% to take into account expected loss to follow-up, the sample size has to increase to 180 patients.

### Recruitment

All eligible children aged 6–12 years in the participating GP practices will be invited to participate. Eligible patients will be selected by searching the electronic patient

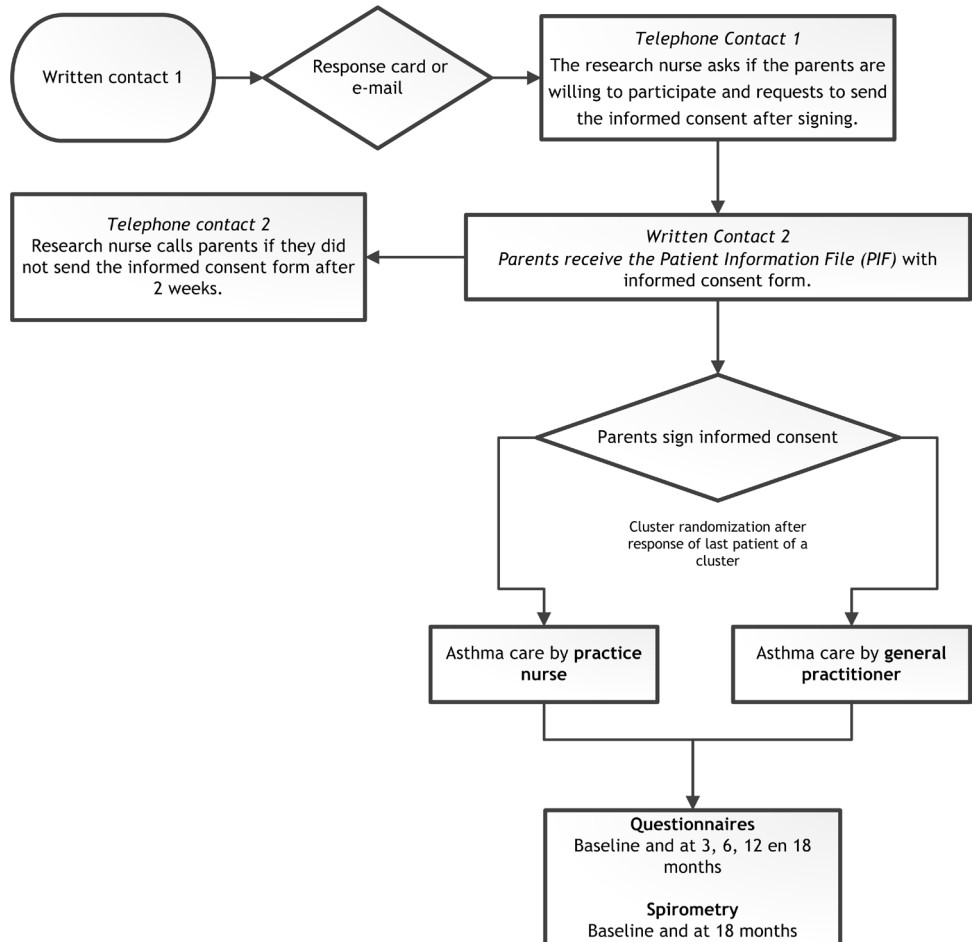

**Figure 3** Flowchart of study procedures. PIF, patient information file.

database of the GPs and applying the inclusion/exclusion criteria. Parents will be asked to return a response form.

### Study procedures (cluster randomised group)

A researcher (nurse or medical student) will telephone the parents to check whether the children are eligible. Parents will receive the patient information letter with the informed consent form by regular post. When the patient and parents are willing to participate, the informed consent will be signed by the parents/legal guardians and sent to the department of General Practice by post. After all the parents of eligible children of one cluster (who are willing to participate) have provided informed consent, the randomisation is conducted by a researcher and the parents are informed (by telephone) in which treatment arm their child is allocated. Children who turn 12 years during the trial also have to give informed consent. The deadline for randomisation is 8 weeks after inclusion of the first patient of a cluster. Figure 3 is a flowchart of the study procedures.

### Study procedures (non-randomised group)

Besides the cluster randomised group of children we will include a non-randomised group of children in this trial. These children will be recruited in health centres who intend to start protocolled nurse-led care for children with

asthma in their practice. Practice nurses of the non-randomised participating GP practices will attend the same training on protocolled asthma care. All eligible children who receive protocolled nurse-led care will be invited to participate in the non-randomised part of this trial. They will be invited by their GP with a slightly adjusted invitation letter, the other study procedures will be identical.

## DATA COLLECTION

### Spirometry

A spirometry test will be conducted at the children's homes at baseline and at 18-month follow-up using a portable device (CareFusion Vyntus Spiro with Sentry-Suite Software V.2.21). The spirometry will be conducted by two specialised nurses to minimise interobserver bias and will be performed according to the guideline of the Dutch College of General Practitioners.[6] The nurses who conduct the spirometry tests will not participate in the protocolled care; they will be appointed as independent research employees. Reversibility will be determined by administering salbutamol via an aerochamber. After the spirometry, the nurse will determine whether the test was properly conducted. Second, the predicted value of FEV1 for age, height and gender will be calculated by a tool of

the European Respiratory Society. A score of −2 (SD) is considered as deviant. If the result of the spirometry indicates less than or equal to −2 SD of FEV1 and the child or parents report symptoms such as wheezing or shortness of breath, the nurse advises parents to inform the GP of the child about the current status of the child. In all other cases, to prevent bias, neither the GP nor the parents or the child will be informed about the test result.

### Baseline measurements and questionnaires

Height and weight, asthma,control according to GINA guidelines, ethnicity of the child, smoking status of the parents and smoking status of the mother during pregnancy will be determined during the baseline visit by the practice nurse. All these items are entered in an online data management system with a secured 'cloud' (Research Manager, Deventer, the Netherlands).

The other questionnaires: the C-ACT, CHU9D, EQ-5D-Y, the questionnaire on satisfaction with delivered care, the questionnaire on the frequency/severity of exacerbations and on healthcare costs will be sent to the parents (digitally) at baseline and at 3, 6, 12 and 18 months. The PAQLQ questionnaire will be sent to the parents only at baseline and at 6 and 18 months.

### Retrospective chart review

Direct and indirect healthcare costs and information on co-morbidity will be assessed at 18 months by retrospective chart review. In the Netherlands, all hospital visits (both emergency room visits and cases that are admitted to stay in hospital) are reported to the GP; this includes the duration of stay. The following items will be assessed: referrals from primary to secondary care, number of hospital admissions, medication costs, number and duration of consultations and consultations with the practice nurse and/or the physician (for asthma and non-asthma related symptoms), evidence of stepping-down therapy (ie, reduction in prescribed medication) and information on allergies.

### Withdrawal of individual participants

Participants can leave the study at any time for any reason if they wish to do so, without any consequences. The investigator can decide to withdraw an individual from the study for any urgent medical reasons.

### Analyses

The results of spirometry, the questionnaires and data from the retrospective chart review will be entered in an online and secured data management system (Research Manager). After merging these files, the data will be processed with IBM SPSS for Windows software.

Baseline characteristics will be summarised using simple descriptive statistics. All analyses will be performed according to the intention-to-treat principle. A per-protocol analysis will also be conducted in patients with a maximum 'no show' percentage of 33% (1/3 of the review visits).

### Primary outcome measures

C-ACT scores will be analysed in a multilevel repeated measures regression model, which incorporates the C-ACT scores for all measurements, including the baseline measurement. This method takes into account within-patient correlation by modelling the covariance structures of the residuals per measurement. Several structures will be considered, starting with the unstructured covariance structure.

The differences in C-ACT scores between the two groups will be modelled by the interaction of treatment and measurement after baseline, assuming that any difference in C-ACT at baseline is due to chance. Additional variables will be considered for inclusion in the regression model if they are substantially different at baseline or if they are expected to be predictive of drop-out at a later stage.

Furthermore, a subgroup analysis will be conducted in which the total sample is divided into three groups according to their baseline characteristics; a group with insufficient asthma control (undertreated children), a group with sufficient asthma control and continuous ICS use (possible overtreated children) use and a group with the remaining children.

### Secondary outcome measurements

Differences in C-ACT scores within a study subject (subjects in the non-randomised group of protocolled nurse led care), will be estimated with a paired Z-test or paired t-test. Besides we will determine differences in outcome between the randomised and non-randomised protocolled nurse-led care samples and the usual care group by using unpaired t-test in case of normally distributed values and Mann-Whitney U test in case of non-normally distributed values. For categorical outcomes, we will use Chi-square test for statistical analyses.

### Missing data

The multilevel regression model also deals with missing observations of the outcome measure by exploiting the fact that observations within patients are correlated, if the missingness occurs completely at random or at random (conditional on the observed data in the model). This allows unbiased estimation of regression coefficients and to make optimal use of all available data, even when some measurements are missing.[32] Dropouts or missing visits will be reviewed to identify possible informative missingness.

### Secondary outcome measures

The numbers of exacerbations will be analysed in Poisson-type regression models, with the time spent in the study as offset, thus adjusting for potential loss to follow-up. The other secondary outcome measures, that is, generic quality of life (measured by CHU9D and EQ-5D), disease-specific quality of life (PAQLQ), satisfaction and 3-monthly costs will be analysed in separate multilevel regression models. The probability of achieving asthma

control at any moment in time will be analysed in a multilevel logistic regression model. Asthma control is defined as a C-ACT score of 20 or better.

### Economic evaluation

A prospective economic evaluation will be performed from both a healthcare perspective and a societal perspective alongside the clinical trial, according to Dutch and international guidelines.[33 34] If nurse-led asthma care leads to more costs, as well as health gains compared with usual care, the cost-effectiveness will be expressed in the following incremental cost-effectiveness ratios (ICERs):

► Costs per quality-adjusted life year (QALY).
► Costs per additional patient with asthma control.

The uncertainty around the estimates of incremental costs and effects, and around the ICERs, will be displayed on cost-effective planes and in cost-effectiveness acceptability curves.

Costs are calculated by measuring all relevant cost categories per interval at the patient level and multiplying natural units by standard prices. The costs are then analysed in a multilevel repeated measures regression model. Based on the regression results, adjusted average costs per interval and for the total study period are calculated for each treatment option. The healthcare perspective includes all medical costs (GP and nurse contacts, hospital admissions, pulmonologist or paediatrician consultations, medication). The societal perspective also includes the costs of parent's production losses as a consequence of their child's asthma.

The QALY calculations are based on the results of the regression analysis of the generic quality-of-life measurements (EQ-5D-Y and CHU-9D). The results are used to calculate adjusted average quality of life at the end of each interval for each treatment. After this, QALYs per interval can be calculated by taking the average of two measurements around the interval and multiplying this by the duration.

The results of a multilevel logistic regression analysis will be used to calculate individual and average probabilities of achieving asthma control at t=18 for each treatment. The costs per additional patient with asthma control can then be calculated as the ratio of the incremental costs and the difference of the average probabilities.

### SAFETY REPORTING
#### Monitoring

In accordance with the Dutch Act on Medical Research Involving Human Subjects (the WMO; section 10, subsection 4), the sponsor will suspend the study when there is sufficient ground that continuation of the study will jeopardise a participant's health or safety. The sponsor will notify the accredited Medical Ethical committee (MEC) without undue delay of a temporary halt, including the reason for such an action. The study will be suspended pending a further positive decision by the accredited MEC. The investigator will ensure that all participants are kept adequately informed.

Due to the characteristics of this study, it is not necessary to install a Data Safety Monitoring Board. Nevertheless, the study will be monitored as described in the ICH-GCP Guidelines (Chapter 5.18). The department of general practice has developed a monitoring plan and a monitoring checklist (based on the ICH-GCP Guidelines) which will be used during this study.

### Adverse events

Serious adverse events (SAEs) will probably be first observed by the treating GP. A serious adverse event is any undesired medical occurrence that is lethal, life-threatening, requires hospitalisation or prolongation of existing inpatients' hospitalisation, results in persistent or significant disability or incapacity, results in a congenital anomaly or birth defect or a undesired medical occurrence which was adequately diverted before serious harm occurred.

The GP is asked to report any SAE to the investigator. In addition, patients will be asked if they have been admitted to hospital when they answer the questionnaires every 3–6 months. Patients will be contacted by the researcher if they have been admitted to hospital. The investigators will process the SAE and report it through a web portal and to the accredited MEC.

### Amendments

Amendments are changes made to the research protocol after a favourable opinion from the accredited MEC. All substantial amendments will be notified to the MEC and to the competent authority.

### Ethics and dissemination

All data will be obtained, managed and monitored according to the guidelines of good clinical practice. This study was approved by the Medical Ethics committee of the Erasmus Medical Center Rotterdam, the Netherlands (MEC-2017–566). Findings of the study will be published in international journals and presented at international conferences. Where appropriate, the results of this study can be directly implemented in the revisions of the guidelines. The researchers are in close contact with policymakers, the Dutch GP-guideline organisation and local organisations that are willing to implement the intervention on a large scale when proven effective.

### Patient involvement

We created will created a panel of two parents of children with asthma. These parents were involved in conducting the patient information brochure concerning the study. They were also asked to assess the design of the study and the burden of the intervention for their children and family.

### DISCUSSION

The aim of this study is to determine whether protocolled practice nurse-led asthma care for children aged 6–12 years in primary care provides more effective asthma control than usual care. This age category was selected because an

asthma diagnosis is difficult to confirm in children aged ≤6 years. In ≥50% of preschool children who experience at least one episode of wheezing before age 3 years, the symptoms will not persist and they will not develop asthma later in life.[6] Moreover, we think that children aged 6–12 years are more willing to follow the advice of their parents, so that they may derive more benefit from the intervention than older, more independent, teenagers. With the introduction of individual care plans and education about pathophysiology, prevention and medication, children and their parents may learn to influence their chronic disease in a more positive way. This could result in better asthma control and might improve the quality of life of children and their family members. These are the reasons for the selection of children aged 6–12 years for this trial.

Asthma in children is a clinical diagnosis for which no 'gold standard' diagnostic criteria are available. Since different definitions of asthma are reported in national/international guidelines,[6 7 16 17 35] this makes difficult to present a clear definition for enrolment in the present trial. However, all definitions include the presence of symptoms (eg, more than one of: wheeze, breathlessness, chest tightness, cough) and of variable airflow obstruction. Airway inflammation is also a frequent characteristic. However, fractional exhaled nitric oxide (FeNO) and eosinophilia measurements are not of added value in the diagnostic process and follow-up of children with asthma.[7] Therefore, these latter tests are not recommended in the current primary and secondary care guidelines for Dutch GPs.[6 7] Moreover, this would entail conducting additional invasive tests (such as blood sampling) in the children; therefore, FeNO and eosinophilia will not be determined in the present trial.

The guideline of the Dutch College of General Practitioners offers the following definition: 'asthma is a clinical diagnosis that may be supported by the results of a spirometry test.[6] Wheezing is the core symptom of asthma'.[6] The selection of eligible children is based on the registered diagnosis in the patient file; International Classification of Primary Care (ICPC) code and/or the use of medication in the previous year. However, this method could result in the selection of children who do not use the medication, despite the prescription. Nevertheless, a registered diagnosis, also referred to as 'physician-diagnosed asthma' has been used as an inclusion criterium in other trials.[11 36 37] For our study, we considered asthma to be present in the children, when they were prescribed an ICS (≥one time) or salbutamol or terbutaline (≥two times) in the previous year or had only one prescription of salbutamol or terbutaline in the previous year and a registered ICPC code for asthma (R96) or R29.02 'Prikkelbare luchtwegen' (bronchial spasm) . We think this is a sufficient indication for the presence of asthma.

Some questionnaires have been validated to determine asthma control in the paediatric population. For example, the Asthma Control Questionnaire (ACQ)[38] and the C-ACT[26] are frequently used in international trials.[17 26 27 38–41] Both questionnaires show good agreement

as compared with the GINA criteria.[40] The ACQ is a valid instrument for measuring asthma control in children aged 6–16 years. It consists of six symptom questions with a recall of 1 week; however, for children aged 6–10 years, it must be administered by a trained interviewer.[42] In the present trial, the questionnaires will be sent by email, since it is not feasible to arrange face-to-face contacts for all the study measurements.

Furthermore, no validated Dutch version of the interviewer administered ACQ is available. Lastly, we assume that children might prefer the C-ACT, since a visual scale with expressed emotions is shown after the questions to help the child determine the correct answer. Therefore, the ACQ is less appropriate than the C-ACT to determine asthma control in the children in this study.

In the present study, two spirometry tests will be conducted. Reversibility will be tested by administering the SABA 'salbutamol' (inhalation medication). The dosage of salbutamol will be according to the guideline of the Dutch College of General Practitioners.[6] Possible side effects include tremor of the hands, headache, peripheral vasodilatation and an increase of heart rate.[6] However, generally, these symptoms only occur when salbutamol is used in a higher dosage. In this proposed trial, the maximum dosage will never be exceeded. Also, the parents of the child will be asked to cease use of airway dilators before the test; this could lead to a temporary increase of the symptoms of asthma. However, this is a standard procedure for spirometry[6] and, of course, when it is not possible or medically irresponsible to stop inhalation medication at that moment, spirometry will be postponed.

The participating child, parents, practice nurse and GP will be informed about the results of the spirometry only when the FEV1 is less than or equal to −2 SD of predicted value for age, height and gender. In this case of severe obstruction of the lungs, it would be unethical not to inform them and provide the GP the opportunity to improve the medical condition of the child. If the results are shared in other cases, there is a risk of increased healthcare consumption because of more awareness of the condition or worries of the parents or GP. This could result in bias, because extra review visits (especially in the usual care arm) could be planned, which could lead to better asthma control.

The randomisation of the intervention (the practice nurse or group of practice nurses) will be conducted by stratified cluster randomisation. A pragmatic trial compares the effectiveness of an intervention in everyday practice with relatively unselected participants and under flexible conditions. Pragmatic trials have a high external validity due to the relatively unselected patients and are, therefore, helpful to answer the question whether the intervention has additional value in real practice.[43] The aim of the current pragmatic trial is to compare protocolled nurse-led care by the practice nurse to the usual care of the GP. Cluster randomised trial designs are common in pragmatic research.[44–46] Cluster randomised trials have particular utility in effectiveness and implementation studies. This design is appropriate for the

evaluation of interventions that are naturally applied at cluster level, for example, general practices that opted for a certain stepped approach. Moreover, cluster randomisation avoids group contamination. Disadvantages of pragmatic cluster randomised trials are the generalisability and the recruitment bias; however, this can be (partly) solved by inviting all eligible patients from a cluster to participate in the trial. Another issue with cluster randomisation is the possible baseline imbalance between the randomised groups; however, this bias can be minimised by stratification of the clusters.

**Acknowledgements** We acknowledge Ms N Maatje and Ms B van Eijk for their involvement in the patient panel of this trial.

**Contributors** AJV, PJEB and GE conceived the idea and secured funding. Ethics applications were made by SB in collaboration with GE, AJV and PJEB. LMG and AMB provided intellectual input concerning study design and statistical analysis. All authors were involved in drafting or critically revising this work, and in final approval of the version to be published.

**Funding** This work was supported by The Netherlands Organisation for Health Research and Development (ZonMw) grant number [839110021].

**Competing interests** None declared.

**Patient consent for publication** Not required.

**Provenance and peer review** Not commissioned; externally peer reviewed.

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
