## [Reviewer comments · BMJ Open]

ARTICLE DETAILS

TITLE (PROVISIONAL)	Protocolled practice nurse-led care for children with asthma in primary care: protocol for a cluster randomised trial
AUTHORS	Bousema, Sara; Verwoerd, Annemieke; Goossens, Lucas; Bohnen, Arthur; Bindels, Patrick; Elshout, Gijs

VERSION 1 – REVIEW

REVIEWER	Andrew Cave Department of Family Medicine, University of Alberta, Canada
REVIEW RETURNED	30-Apr-2018

GENERAL COMMENTS	This study is not new Nurse-run asthma clinics have been the norm in the UK since the 1970s and there have been several evaluations by the Stratford based Asthma Centre. We performed one in adults/paeds in 1991 (Cave et al PCRJ 2001). Asthma prevalence in the Netherlands seems low at 6% and the physician based diagnosis needs better definition(? a simple mention without symptoms or spirometry, multiple prescriptions of asthma meds, hospitalization?) The aims vary slightly but importantly from abstract to page 6 line 20 and back to primary measure on page 7 line 24. The inclusion /exclusion criteria need better defining . It is a shame that children with other comorbidities will be excluded."average" asthma control needs defining. As the authors say C-QACT has a 4 week window and may not be the best tool for the study of a condition that can vary so much in a month. How about a tool that measures symptoms over a week. See the IPCRG guidelines for choice of tools for asthma. Secondary outcomes (2) asthma episodes may be rapid as well as 'progressive'. hospital visits and so costs may not be contained in the GP chart. How will the authors handle this? To whom is the questionnaire going and to ask what? Follow up visits (page 8) Are these protocol or need based visits? Will the patient carry the Action plan? The authors seem to believe that good control means that the patient can have the medication reduced as 'overtreated' but it may be 'ideally treated'. To avoid contamination within practices (some patients with one nurse will be intervention group and those with another nurse could be 'control' patients) it would be better to randomize by practice don't you think? Will you need block randomization with only 23 practices unless there is slow recruitment of practices when you want to get started on the first four to agree . this avoids starting all the practices on the same day (very difficult). Also , I do
--

	not think you can expect all the children in a cluster to consent Page 10 line 56) The baseline spirometry should be off treatment. a good idea to do outside the clinics and to have independent nurses perform it. There are a lot of questionnaires. Do you need them all? Will hospitalization be consistently recorded in the GP chart?(page 12 line 1). Will the nurse also prescribe (line2) As is common, the process of economic evaluation is inadequately described and so I do not think the authors know how to do it and what is involved. Serious Adverse Events need better definition.(page 14) The idea that the results will be directly implemented into the guidelines is fanciful thinking, not a dissemination plan. The discussion says that the AIM is to optimize management (page 15 line 4) but that is NOT the aim of the study though it might be the underlying intention. The AIM is in the abstract.. The second half of this first paragraph (page 15) is very weak and hopeful. I am still not convinced that symptoms alone such as wheezing can define asthma. Flow diagrams are a helpful addition as are the supplementart documents. As mentioned , the literature could be reviewed.
--	--

REVIEWER	Maciek Godycki-Cwirko, professor Medical University of Lodz. Poland
REVIEW RETURNED	21-Jun-2018

GENERAL COMMENTS	Well chosen, relevant research question. Well designed and elaborated protocol. Looking forward to results
--

REVIEWER	Fan Li Department of Biostatistics and Bioinformatics, Duke University Medical Center, USA
REVIEW RETURNED	16-Jul-2018

GENERAL COMMENTS	A cluster randomized trial (CRT) is planned by the authors' team to determine whether practice nurse-led asthma care for children provides more effective asthma control than regular care. The detailed study protocol is presented in the manuscript. Given the important public health relevance of the study, careful statistical design and analysis are required to inform future practice for improved asthma control. I have several clarification questions on the statistical part of the protocol, which should be addressed before publication. 1. (Randomization) In the terminology of CRTs, a cluster is usually the unit of randomization. On page 10-11, the authors indicated that each nurse will be the unit of randomization, which should then be a "cluster". But nurses are nested in GP practices, and the authors frequently referred to a practice as a cluster. Is it true that each GP practice may have multiple nurses, and nurses are nested within GPs, and so the current study is actually a three-level cluster randomized trial? If so, could you clarify whether the randomization is conducted at the nurse-level or GP level?
---

	2. (Randomization) When stating “block randomization”, did you mean stratified randomization with strata defined by the same values of the stratification variables (size and experience)? So each block is a stratum? Some consistency in terminology is desired to avoid confusion on the study design. Stratified randomization or stratification is the standard language in CRTs, please see [1]. [1] Turner EL, Li F, Gallis J, Prague M, Murray DM (2017). Review of recent methodological developments in group-randomized trials: Part 1–Design. American Journal of Public Health, 107 (6), 907-915. 3. (Primary outcome measures) When using the multilevel regression model to analyze the in C-ACT scores, are you specifying baseline measurements as well as all follow-up measurements as outcomes? If so, could you state clearly what random effects do you plan to control for? Since there are repeated measurements, you may need consider an individual-specific random effect to account for the correlation between repeated assessments. Also, relating to my comment 1, if it is a three-level CRT, then a GP-level random effect and a nurse-level random effect may also be required. 4. (Missing data) The statement “The multilevel regression model also deals with missing observations of the outcome measure by exploiting the fact that observations within patients are correlated” may be misleading. The validity of multilevel regression with missing data is grounded by the theory of likelihood inference, and the “missing at random” assumption. When there is informative missingness, i.e., drop-out or missed visits are due to worse prognosis, multilevel regression models may have biased estimates. The author may clarify that additional review of drop-out or missing visits may be required to assess the possibility of informative missingness.
--	--

VERSION 1 – AUTHOR RESPONSE

Reviewer: 1

Reviewer Name: Andrew Cave

Institution and Country: Department of Family Medicine, University of Alberta, Canada Please state any competing interests or state 'None declared': None declared

Please leave your comments for the authors below

This study is not new Nurse-run asthma clinics have been the norm in the UK since the 1970s and there have been several evaluations by the Stratford based Asthma Centre. We performed one in adults/paeds in 1991 (Cave et al PCRJ 2001).

Authors' response:

We disagree with reviewer 1 that the study is not new. Asthma run clinics are not new, but randomized trials on nurse-led asthma care in primary care is scarce until now (see also the reference the reviewer added: Cave et al PCRJ 2001). It is shown by a systematic review that there is limited evidence of efficacy for primary care based asthma clinics, and firm conclusions cannot be formed until more good quality trials have been carried out (Baishnab et al. Cochrane Database Syst Rev 2012). We have added this in the manuscript (line 156).

Reviewer comments:

Asthma prevalence in the Netherlands seems low at 6% and the physician based diagnosis needs better definition(? a simple mention without symptoms or spirometry, multiple prescriptions of asthma meds, hospitalization?)

Authors' response:

In this trial, asthma is defined not only by the diagnosis once recorded by the GP, but also on the use of medication for asthma in the last year. This was based on the work of Pols et al (Pols DHJ, Nielen MMJ, Korevaar JC, et al. Reliably estimating prevalences of atopic children: an epidemiological study in an extensive and representative primary care database. NPJ Prim Care Respir Med 2017;27(1):23.) and prevents the inclusion of cases of asthma that are for many years without symptoms or misdiagnosed in the past at a young age. This may explain the relatively low prevalence of 6% as reported by Pols.

The physician based diagnoses is reflected in our inclusion criteria (line 185):

- Patients who were prescribed an ICS one or more times in the previous year.
- Patients who were prescribed salbutamol or terbutaline two or more times in the previous year.
- Children with only one prescription of salbutamol or terbutaline in the previous year and a registered International Classification of Primary Care code for asthma (R96).

Reviewer comments:

The aims vary slightly but importantly from abstract to page 6 line 20 and back to primary measure on page 7 line 24.

Authors' response:

We clarified the aim of the study: Primary outcome will be average asthma control during the 18-month follow-up measured by the Childhood Asthma Control Test (C-ACT) (line 49).

The aim of this study is to determine the effectiveness of protocolled nurse-led asthma care for children aged 6-12 years in primary care as compared to usual care. The overall aim is to improve asthma care for children in primary care. This will be measured by determining the average asthma

control during the 18-month follow-up, as measured by the Childhood Asthma Control Test (C-ACT). Besides improvement for individual children, this may lead to a reduction of the social burden of the disease. Physician-nurse substitution, with a special focus on 'proactive care', may reduce healthcare costs whilst maintaining quality of care (line 163-169).

Reviewer comments:

The inclusion /exclusion criteria need better defining. It is a shame that children with other comorbidities will be excluded.

Authors' response:

By 'significant comorbidities', we mean that children with major chronic comorbidities that are treated by a pediatrician in secondary care, for whom it will be more practical to receive asthma care by the pediatrician as well. We clarified this in line 196 (Children with other major chronic diseases which are treated in secondary care).

Reviewer comments:

"average" asthma control needs defining.

Authors' response:

We mean the average C-ACT score of the measurements on baseline, and 3, 6, 12 and 18 months. We clarified this in line 208 (The average C-ACT score will be determined using the measurements on baseline, and 3, 6, 12 and 18 months.)

Reviewer comments:

As the authors say C-QACT has a 4 week window and may not be the best tool for the study of a condition that can vary so much in a month. How about a tool that measures symptoms over a week. See the IPCRG guidelines for choice of tools for asthma.

Authors' response:

Since we measure asthma control only at baseline and 4 times thereafter, we prefer a tool that measures asthma control over a 4 week period above a tool that measures over a period of one week, thus capturing some of the variability that is mentioned by the reviewer.

Reviewer comments:

Secondary outcomes

(2) asthma episodes may be rapid as well as 'progressive'. hospital visits and so costs may not be contained in the GP chart. How will the authors handle this? To whom is the questionnaire going and to ask what?

Authors' response:

Asthma episodes may be rapid and result in hospital visits. In the Netherlands all hospital visits (both emergency room visits as well as cases that are admitted to stay in hospital) are reported to the GP, this includes the duration of stay. At the end of the study, the patient charts of the GP are studied to find both visits to the GP practice as well as visits to the hospital. We clarified this in line 376 (In the Netherlands all hospital visits (both emergency room visits as well as cases that are admitted to stay in hospital) are reported to the GP, this includes the duration of stay.)

The questionnaires are sent to the parents of the patients. This clarified in line 222.

Reviewer comments:

Follow up visits (page 8) Are these protocol or need based visits? Will the patient carry the Action plan?

Authors' response:

The frequency of these visits will be determined by the protocol in the intervention group, in which the frequency is increased in case of poor asthma control or in case of changes in medication. This is described in line 257-272).

Reviewer comments:

The authors seem to believe that good control means that the patient can have the medication reduced as 'overtreated' but it may be 'ideally treated'.

Authors' response:

We believe that ideally treated means a minimum of symptoms with the lowest possible dose of medication. It is known that overtreatment occurs in children with asthma (line 105-107).

Reviewer comments:

To avoid contamination within practices (some patients with one nurse will be intervention group and those with another nurse could be 'control' patients) it would be better to randomize by practice don't you think? Will you need block randomization with only 23 practices unless there is slow recruitment of practices when you want to get started on the first four to agree . this avoids starting all the practices on the same day (very difficult).Also , I do not think you can expect all the children in a cluster to consent Page 10 line 56)

Authors' response:

Several GP-practice may be located in one health center, in which one or more practice nurses may be working. In such cases, the whole center is randomized to either the intervention or control group, and considered as one cluster. We clarified this in the section 'Randomisation' (line 297: The participating health centers (i.e. one or more collaborating GP practices) will be the units of randomisation. They will be randomly assigned to either the intervention or control group with a 1:1 allocation as per a computer-generated stratified randomisation schedule. When a practice nurse works for several GP practices, all these practices will be part of the same cluster and randomized as one group. When a health centre consists of several GP practices with their own practice nurse, this health centre will also be considered as one cluster. When two practice nurses are working in one GP practice (not separated), this GP practice or health centre will be randomised as one cluster).

The cluster randomization is done by a computer program that allows block randomization one by one. We stop inclusion of children in a specific practice if in four weeks no further children sign up.

Reviewer comments:

The baseline spirometry should be off treatment. a good idea to do outside the clinics and to have independent nurses perform it.

Authors' response:

In this case, it is not necessary to perform the baseline spirometry off treatment. We do not need to determine asthma severity without medication, but rather the additional value of nurse-led care as compared to usual care.

Reviewer comments:

There are a lot of questionnaires. Do you need them all?

Authors' response

We administer a substantial amount of questionnaires. However, they are only administered 5 times in 18 months, so we think this is feasible for the patients.

Reviewer comments:

Will hospitalization be consistently recorded in the GP chart?(page 12 line 1).

Authors' response:

In the Netherlands all hospital visits (both emergency room visits as well as cases that are admitted to stay in hospital) are reported to the GP, this includes the duration of stay. At the end of the study, the files of the GP are studied to find both visits to the GP practice as well as visits to the hospital. We clarified this in line 374 (In the Netherlands all hospital visits (both emergency room visits as well as cases that are admitted to stay in hospital) are reported to the GP, this includes the duration of stay.)

Reviewer comments:

Will the nurse also prescribe (line2).

Authors' response:

In the Netherlands, nurses are not allowed to prescribe without final approval by the GP. In daily practice they will contact the GP in case of a new or repeat prescription. This is clarified in line 248 (When new or repeated medication is needed, the GP will approve the prescription.)

Reviewer comments:

As is common, the process of economic evaluation is inadequately described and so I do not think the authors know how to do it and what is involved.

Authors' response:

Unfortunately, the reviewer did not make clear what he sees as the inadequacy of the description of the process of the economic evaluation and how this led to his belief that the authors do not know what is involved in an economic evaluation. That makes it difficult for us to respond to this comment. An experienced health care economist is involved in the economic evaluation (Lucas Goossens, PhD) and the new version of the manuscript stresses that we will follow Dutch and international guidelines for economic evaluations in healthcare. Furthermore, we have extended the description of process in the manuscript, with a clearer distinction between the regression analyses and the subsequent calculation of the outcome measures for the economic evaluation (line 436, 446-450, 454-465).

Reviewer comments:

Serious Adverse Events need better definition.(page 14)

Authors' response:

We added to the manuscript in line 482-486: A serious adverse event is any undesired medical occurrence that is lethal, life threatening, requires hospitalization or prolongation of existing inpatients' hospitalization, results in persistent or significant disability or incapacity, results in a congenital anomaly or birth defect, or a undesired medical occurrence which was adequately diverted before serious harm occurred.

Reviewer comments:

The idea that the results will be directly implemented into the guidelines is fanciful thinking, not a dissemination plan.

Authors' response:

We added to the dissemination plan in line 503: the researchers are in close contact with policy makers, the Dutch GP-guideline organization, and local organizations that are willing to implement the intervention on a large scale when proven effective.

Reviewer comments:

The discussion says that the AIM is to optimize management (page 15 line 4) but that is NOT the aim of the study though it might be the underlying intention. The AIM is in the abstract.. The second half of this first paragraph (page 15) is very weak and hopeful.

I am still not convinced that symptoms alone such as wheezing can define asthma.

Authors' response:

We changed the first sentence of the discussion. Line 515: The aim of this study is to determine whether protocolled practice nurse-led asthma care for children aged 6-12 years in primary care provides more effective asthma control than usual care.

Concerning the asthma definition, we added in line 548 'For our study, we considered asthma to be present in the children, when they were prescribed an ICS (>one time), or salbutamol or terbutaline (>two times) in the previous year, or had only one prescription of salbutamol or terbutaline in the previous year and a registered International Classification of Primary Care code for asthma (R96). We think this is a sufficient indication for the presence of asthma.'

Reviewer comments:

Flow diagrams are a helpful addition as are the supplementart documents.

As mentioned , the literature could be reviewed.

Authors' response:

As described above, It is shown by a systematic review that there is limited evidence of efficacy for primary care based asthma clinics, and firm conclusions cannot be formed until more good quality trials have been carried out (Baishnab et al. Cochrane Database Syst Rev 2012). We have added this in the manuscript (line 156).

Reviewer: 2

Reviewer Name: Maciek Godycki-Cwirko, professor Institution and Country: Medical University of Lodz. Poland Please state any competing interests or state 'None declared': None

Please leave your comments for the authors below

Well chosen, relevant research question.

Well designed and elaborated protocol.

Looking forward to results

Authors' response:

We thank the reviewer for his positive evaluation of our study and submitted manuscript.

Reviewer: 3

Reviewer Name: Fan Li

Institution and Country: Department of Biostatistics and Bioinformatics, Duke University Medical Center, USA Please state any competing interests or state 'None declared': None declared

Please leave your comments for the authors below

A cluster randomized trial (CRT) is planned by the authors' team to determine whether practice nurse-led asthma care for children provides more effective asthma control than regular care. The detailed study protocol is presented in the manuscript. Given the important public health relevance of the study, careful statistical design and analysis are required to inform future practice for improved

asthma control. I have several clarification questions on the statistical part of the protocol, which should be addressed before publication.

1. (Randomization) In the terminology of CRTs, a cluster is usually the unit of randomization. On page 10-11, the authors indicated that each nurse will be the unit of randomization, which should then be a “cluster”. But nurses are nested in GP practices, and the authors frequently referred to a practice as a cluster. Is it true that each GP practice may have multiple nurses, and nurses are nested within GPs, and so the current study is actually a three-level cluster randomized trial? If so, could you clarify whether the randomization is conducted at the nurse-level or GP level?

Authors’ response:

Several GP-practice may be settled in one health center, in which one or more practice nurses may be working. In such cases, the whole center is randomized to either the intervention or control group, and considered as one cluster. We clarified this in the section ‘Randomisation’ (line 292: The participating health centers (i.e. one or more collaborating GP practices) will be the units of randomisation. They will be randomly assigned to either the intervention or control group with a 1:1 allocation as per a computer-generated stratified randomisation schedule. When a practice nurse works for several GP practices, all these practices will be part of the same cluster and randomized as one group. When a health centre consists of several GP practices with their own practice nurse, this health centre will also be considered as one cluster. When two practice nurses are working in one GP practice (not separated), this GP practice or health centre will be randomised as one cluster).

Reviewer comments:

2. (Randomization) When stating “block randomization”, did you mean stratified randomization with strata defined by the same values of the stratification variables (size and experience)? So each block is a stratum? Some consistency in terminology is desired to avoid confusion on the study design. Stratified randomization or stratification is the standard language in CRTs, please see [1].
[1] Turner EL, Li F, Gallis J, Prague M, Murray DM (2017). Review of recent methodological developments in group-randomized trials: Part 1–Design. *American Journal of Public Health*, 107 (6), 907-915.

Authors’ response:

1. We mean it as described by the reviewer and added this to the paragraph (line 312, For example, GP practices in a single health centre (the same postal code) will be randomised in a stratified randomisation of four groups and, therefore, evenly distributed to the intervention and the control arm to minimise differences in baseline characteristics of the patients.).

Reviewer comments:

3. (Primary outcome measures) When using the multilevel regression model to analyze the in C-ACT scores, are you specifying baseline measurements as well as all follow-up measurements as outcomes? If so, could you state clearly what random effects do you plan to control for? Since there are repeated measurements, you may need consider an individual-specific random effect to account for the correlation between repeated assessments. Also, relating to my comment 1, if it is a three-level CRT, then a GP-level random effect and a nurse-level random effect may also be required.

Authors’ response:

It is a two level RCT (patients are nested within nurses, nurse = practice). The text about the statistical analysis has been clarified as follows (line 401):

C-ACT scores will be analysed in a multilevel repeated measures regression model, which incorporates the C-ACT scores for all measurements, including the baseline measurement. This method takes into account within-patient correlation by modelling the covariance structures of the

residuals per measurement. Several structures will be considered, starting with the unstructured covariance structure.

The differences in C-ACT scores between the two groups will be modelled by the interaction of treatment and measurement after baseline, assuming that any difference in C-ACT at baseline is due to chance. Additional variables will be considered for inclusion in the regression model if they are substantially different at baseline or if they are expected to be predictive of drop-out at a later stage.

Reviewer comments:

4. (Missing data) The statement “The multilevel regression model also deals with missing observations of the outcome measure by exploiting the fact that observations within patients are correlated” may be misleading. The validity of multilevel regression with missing data is grounded by the theory of likelihood inference, and the “missing at random” assumption. When there is informative missingness, i.e., drop-out or missed visits are due to worse prognosis, multilevel regression models may have biased estimates. The author may clarify that additional review of drop-out or missing visits may be required to assess the possibility of informative missingness.

Authors’ response:

We added the qualification that the validity is based on the missingness following a MAR or MCAR pattern (line 420). We will review drop-outs or missing visits to identify possible informative missingness. We have added this in line 424 (Drop-outs or missing visits will be reviewed to identify possible informative missingness.)

VERSION 2 – REVIEW

REVIEWER	Fan Li Department of Biostatistics and Bioinformatics, Duke University Medical Center, USA
REVIEW RETURNED	03-Dec-2018
GENERAL COMMENTS	The authors have addressed my concerns and comments. I am looking forward to seeing the results of the study.

VERSION 2 – AUTHOR RESPONSE

Reviewer: 3

Reviewer Name: Fan Li. Institution and Country: Department of Biostatistics and Bioinformatics, Duke University Medical Center, USA

Please state any competing interests or state ‘None declared’: None declared

Authors’ response:

We have stated 'None declared'.

We hope you will find our revised manuscript suitable for publication.\